# Stochastic representation decision theory: How probabilities and values are entangled dual characteristics in cognitive processes

**Giuseppe M. Ferro** [1]*, **Didier Sornette**[1,2,3,4]

**1** Department of Management, Technology and Economics, ETH Zürich, Zürich, Switzerland, **2** Institute of Risk Analysis, Prediction and Management, Academy for Advanced Interdisciplinary Studies, Southern University of Science and Technology, Shenzhen, China, **3** Tokyo Tech World Research Hub Initiative (WRHI), Institute of Innovative Research, Tokyo Institute of Technology, Tokyo, Japan, **4** Swiss Finance Institute, University of Geneva, Geneva, Switzerland

⊚ These authors contributed equally to this work.
* ferrog@ethz.ch

**Data Availability Statement:** All relevant data are within the manuscript and its Supporting Information files.

**Funding:** The authors received no specific funding for this work.

## Abstract

Humans are notoriously bad at understanding probabilities, exhibiting a host of biases and distortions that are context dependent. This has serious consequences on how we assess risks and make decisions. Several theories have been developed to replace the normative rational expectation theory at the foundation of economics. These approaches essentially assume that (subjective) probabilities weight multiplicatively the utilities of the alternatives offered to the decision maker, although evidence suggest that probability weights and utilities are often not separable in the mind of the decision maker. In this context, we introduce a simple and efficient framework on how to describe the inherently probabilistic human decision-making process, based on a representation of the deliberation activity leading to a choice through stochastic processes, the simplest of which is a random walk. Our model leads naturally to the hypothesis that probabilities and utilities are entangled dual characteristics of the real human decision making process. It predicts the famous fourfold pattern of risk preferences. Through the analysis of choice probabilities, it is possible to identify two previously postulated features of prospect theory: the inverse S-shaped subjective probability as a function of the objective probability and risk-seeking behavior in the loss domain. It also predicts observed violations of stochastic dominance, while it does not when the dominance is "evident". Extending the model to account for human finite deliberation time and the effect of time pressure on choice, it provides other sound predictions: inverse relation between choice probability and response time, preference reversal with time pressure, and an inverse double-S-shaped probability weighting function. Our theory, which offers many more predictions for future tests, has strong implications for psychology, economics and artificial intelligence.

**Competing interests:** The authors have declared that no competing interests exist.

## Introduction

Randomness is a fundamental component in most human affairs, from economics and politics to medicine and sports. Yet, people often make poor and inconsistent decisions when confronted with it. The rational normative recipe of Expected Utility Theory [1] has shown major limitations in accounting for how strongly people misperceive probabilities and uncertainty [2–4], leading to the notion of bounded rationality [5] and a long list of behavioral biases and fallacies. Several attempts [6–12] have been made to explain such fallacies, replacing the objective probabilities of events with "decision weights", but still retaining a sort of expectation principle, where the attractiveness of an event is decomposed into the product of (subjective) probability and (subjective) value. Numerous evidence [13–18] however suggest that the two are not independent; for example, people tend to overestimate the probability of an event if the associated outcome is bad. Rank-Dependent Theories [9–11] partially take into account the effect of value on probability, such that decision makers tend to overweight only events with 'extreme' consequences. However, their axiomatic structure prevents them to account for observed violations of stochastic dominance [19]. Operationally, estimating the subjective probability and utility as two separate entities is subjected to the joint-hypothesis problem [20] leading to severe limitations for real-life applications.

The above-cited frameworks are deterministic in nature, postulating that the best option will always be chosen. When tested against empirical data, a probabilistic component is needed [21] to account for observed "noise" and "inconsistencies" [22, 23]. We can distinguish two classes of probabilistic theories of decision-making: random utility maximization (RUM) models and stochastic decision processes. The former, introduced by Thurstone [24] assumes that the "perceived" utility of an option is a random variable, written as the sum of a "true" fixed utility and a random disturbance, encoding the deviation from rational behaviour. Debreu [25] proves the existence of a utility function representing a stochastic preference relation with minimal set of assumptions. McFadden [26] describes the evolution of RUM models over the past decades, linking it to the Luce choice axiom (LCA) [27], a very useful assumption enforcing desirable properties such as independence from irrelevant alternatives and strong stochastic transitivity. However, several empirical studies [28–33] show how humans do not always conform to such structure, the most famous example being the "red bus/blue bus" problem [34].

The second class of models assumes that the utility of alternatives is fixed, but the process leading to a decision is inherently stochastic. Regarding choice in uncertain environment, the most famous model is decision field theory (DFT) [35], where a stochastic process (Brownian motion) is assumed to mimic the fuzzy and hesitant deliberation activity of human mind. The theory takes inspiration from the Ratcliff Drift-diffusion models (DDMs) [36–38], which have shown to well describe choices and reaction times in perceptual decision-making tasks (for example, discriminating a motion direction). Differently from RUM models, these theories can account for the fact that human decision making happens in finite time, and explain how such deliberation time–as well as time pressure–affects choice probability.

In almost all theories, the "true" utility of a gamble or its index of worth is obtained by combining probabilities and outcomes in a (subjective) expectation, irrespective of the probabilistic model adopted (additive random disturbance or drift-diffusion process). As a result, all these frameworks carry some problematic aspects of expected utility theory, the most prominent (and oldest) being embodied in the St. Petersburg paradox [39], where an infinite expectation value of the gamble would imply infinite willingness to pay, while in reality many people would pay at most a small amount [40]. Although Expected Utility was designed to solve such paradox, a simple modification of the gamble, often called Super St. Petersburg paradox [41],

reintroduces the problem: if the lottery provides outcome $2^{2^n}$ – rather than $2^n$ –with probability $1/_{2^n}$, the Bernoullian expected utility of the gamble (logarithmic utility function) diverges again. For any unbounded utility function, there will always be a Super Super St. Petersburg paradox.

When shifting from the normative perspective of decision theory (telling what people should do)—where expected utility proves best—to a descriptive perspective (reporting what people actually do), it is worthwhile to investigate alternative mechanisms of value formation in human mind that are different from the class of generalized expectation approaches mentioned above. Indeed, from a semantic perspective, separating probability and outcome seems quite odd, since any probabilistic statement *must contain* explicitly or implicitly information about "value". In other words, a probability number quantifies the likelihood of a concrete event that is specified, and this event carries an explicit value or implicit assessment of worth or impact. For instance, when conceiving the likelihood of a natural disaster, one cannot help not thinking of the potential associated destruction and losses of lives, which are therefore implicitly connected to a cost. When thinking of the probability of the election of some political candidate, one cannot avoid envisaging the social, economic and financial consequences, which carry an implicit value judgement. Generally, whatever the event, it carries either a direct value or an indirect value assessment, even if not fully formalized in the mind of the probability assessor. Therefore, the way in which outcomes and probabilities interact in human mind seems to be much more entangled than represented by the simple factorization prevalent in utility theories and their generalizations in behavioral economics and psychology. The intermingled nature of probabilities and values have been reported by Lopes [42] and is highlighted in the above-cited experiments [13–18], which demonstrate the effect of outcomes on perceived probability.

## Our contribution

Here we propose a new framework for describing human decisions under risk, based on a representative stochastic process–in the same spirit of drift-diffusion models–but with a notable difference: outcomes and probabilities are not merely multiplied to form an index of worth, rather they combine in a non-symmetric and non-separable way, as dual characteristics of an event. The difference will be evident when presenting the model into details, but the core concept is the following. In drift-diffusion models, as in DFT and race models, outcomes and associated probabilities of gambles are combined in a unique entity, a mathematical expectation, that then plays the role of a drift component of the stochastic process representing the decision-maker. The decision is triggered when the process reaches a threshold, called decision criterion, usually related to the time available for making a choice (the closer the threshold to the starting point, the faster the process will reach it). In our framework, probabilities and outcomes play a structurally different role; a decision occurs when the diffusive particle is absorbed at the end-point of an interval associated with a given event, whose distance is solely determined by the event's probability. The existence of $n$ events is thus represented by $n$ absorbing end-points at the end of $n$ arms in a starfish configuration along which the Brownian particle diffuses. The $n$ arms have different lengths controlled by the probabilities of the associated events. In this representation, it is natural to conceptualize the values or utilities of the outcomes by adding drifts characterizing each arm of the starfish, the larger the value of an event, the larger the drift that biases the random walk towards the corresponding end-point. Notice that this mapping respects the positivity of the probabilities associated with the arms' lengths, while the drifts can be attractive or repulsive to reflect gains and losses, respectively.

More concretely, consider several outcomes $A,B,C\ldots$, each understood to occur with probabilities $p_A,p_B,p_C,\ldots$ Our key idea is that the mind imagines consciously or unconsciously

some bundles of random paths wandering around in some abstract space, where the alternative outcomes $A,B,C...$ are identified as distinct domains (absorbing boundaries) in this space. The distance between domain representing outcome, say, $A$ and the initial position of the particle is inversely proportional to $p_A$, while the bias responsible for the attraction of the particle to the boundary, is proportional to the outcome $A$. The probability for the diffusing particle to be absorbed by a particular domain is then primarily interpreted as a measure of attractiveness of the associated event, as in DFT; at the same time, a **conditional** absorption probability can be interpreted as a subjective value-distorted probability, as we will see below.

Thanks to the mutual interaction between perceived probabilities and perceived value of outcomes embedded in the starfish geometry with drifts, our model predicts the famous four-fold pattern of risk preferences [43]. To get an intuition on why this is the case, we derive two previously postulated features of prospect theory [43]: the inverse S-shaped subjective probability as a function of the objective probability and risk-seeking behaviour in the loss domain. However, these two entities are not exactly those described by prospect theory, because they are not separable. Rather, they can be inferred and rationalized by studying how the predicted choice probability depends on events' outcomes and probabilities. Without added assumptions, our model conforms naturally to Luce choice axiom [27], enforcing strong stochastic transitivity for pairwise choices. It also predicts violations (as well as observance) of stochastic dominance, in agreement with empirical data [44].

Moreover, generalizing the model to account for time pressure and finite decision times, it provides other empirically confirmed predictions: the inverse relation between choice probability and response time [45], preference reversal through time pressure [46, 47], and an inverse double-S-shaped probability weighting function [48]. Also, note that while usual drift-diffusion models have non-trivial and somehow artificial generalizations beyond binary choices [49], our representation remains essentially locally uni-dimensional for an arbitrary number of available options.

Notwithstanding its predictive power, given its simplicity, the present version of our model has limitations. Because of Luce choice structure: i) it would predict a non-deterministic choice for a decision between two simple sure outcomes (thus we restrict our choice set, as Luce does); ii) it cannot predict observed violations of independence from irrelevant alternatives [31–33] (similarity effect, attraction effect, compromise effect). Furthermore, the proposed stochastic representation is more of an allegory that should not be taken at literally meaning that the human brain imagines all possible random paths wandering around in some abstract space for several outcomes $A$, $B$, $C...$, for instance as a result of limited human working memory. Our framework is proposed as a first minimal complexity model or null-model of human risky choice, which provides the baseline for further elaboration and improvements. Indeed, our present model is characterized essentially by only two tuning parameters (compared for instance to the seven parameters of DFT). In the future, we will present extensions of the model obtained by relaxing some assumptions.

In summary, motivated by: i) empirical evidence for "interaction" between probability and value [13–18]; ii) empirical evidence for intrinsically probabilistic human choice [50]; iii) success of drift-diffusion models in describing human behaviour in several tasks, we present a new probabilistic decision theory that combines probability and value in a non-separable way. Despite its simplicity, it provides straightforward derivations at a more microscopic level of several known structures that have been documented empirically in human decision theory. The rest of the article is structured as follows: in Section *Model* we introduce the theoretical model, first without time constraints and then generalizing. In Section *Results*, we outline the main predictions of our theory. Section *Discussion* summarizes and concludes.

## Model

### Stage 1: "Infinite time" Stochastic Representation Decision Theory (SRDT)

This sub-section presents the simplest version of our model, i.e. without considering the role of (finite) time for human decision-making.

**Formulation of the stochastic representation of lotteries.**   In the simplest possible situation, a decision maker (DM) has to make a choice between playing two binary lotteries:

$$L_1 = \{o_A, p; o_B; 1 - p\} \ \text{ or } \ L_2 = \{o_C, q; o_D; 1 - q\} \tag{1}$$

If the DM chooses lottery $L_1$ (resp. $L_2$), she will receive amount $o_A$ (resp. $o_C$) with probability $p$ (resp. $q$), and $o_B$ (resp. $o_D$) with probability $1-p$ (resp. $1-q$). The amounts can be negative, corresponding to losses.

As mentioned in the introduction, our model is conceptually analogous to drift-diffusion models, including decision field theory (DFT), i.e. a stochastic process is assumed to represent the human deliberation activity leading to a decision; choice is triggered when the process reaches a certain threshold. Fig 1 shows how the above binary choice is represented in DFT: if the process (Brownian particle in the simplest case) reaches the upper boundary (resp. lower boundary) first, then lottery $L_1$ (resp. $L_2$) is chosen. The drift component of the motion is related to the difference between expected-like utilities of the lotteries

$$d = EU(L_1) - EU(L_2)$$
$$EU(L_1) = \pi(p)u(o_A) + \pi(1 - p)u(o_B) \tag{2}$$
$$EU(L_2) = \pi(q)u(o_C) + \pi(1 - q)u(o_D)$$

where $u$ and $\pi$ are the so-called utility and probability functions, respectively.

In our framework, an alternative way of value formation is assumed, keeping in mind the numerous evidence [13–18] showing relevant interaction between probability and value perception. We start from a plausible representation of the lotteries' objective probabilities, as perceived by the decision maker. Typically, humans find easier to understand probabilities in terms of frequencies [51]. Therefore, we propose to model their cognition via the occurrence of favorable random walk paths that hit some target, an absorbing boundary in this case. In other words, we view the cognitive processes leading to the "feeling" or "understanding" of probability as imagining a bundle of random walkers wandering about, and the perception of the actual occurrence of the event as the arrival of random walkers in some boundaries or some domains. This representation allows one to give substance and meaning to what is the perception of probability, equal to the fraction of "successful" paths, in the standard frequentist approach of probability theory [52].

Once the lotteries' objective probabilities are encoded into some absorption probabilities, we introduce lotteries' outcomes and account for: i) their intrinsic utility; ii) their effect on perceived probability. The simplest incarnation of this twofold effect is to introduce an outcome-dependent force (derived from a potential energy) that biases the random walk, producing a value-distorted understanding of probability. This construction leads to an effective influence between probabilities and outcomes; such reciprocal interaction will result in a distorted perception of these two entities by the decision-maker, that in turn determines her decision preferences.

Put differently, instead of compressing all the lottery information into an expectation-like index of worth, we "unpack" a lottery by introducing an absorbing branch for each of its outcome-probability pairs. As a consequence, the **topology** of the space where the stochastic

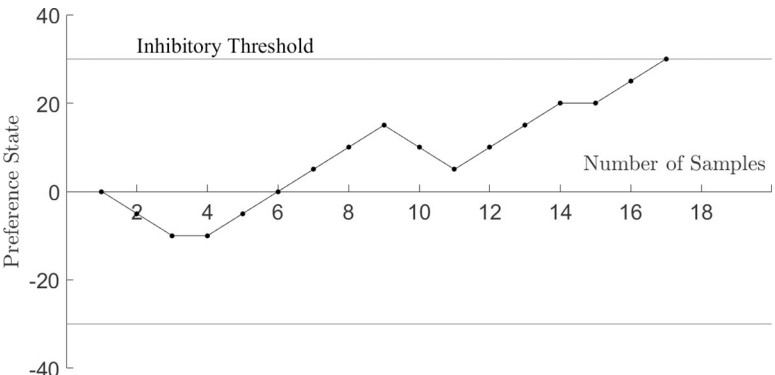

**Fig 1. DFT-representation of binary choice.** If the process reaches the upper boundary (resp. lower boundary) first, then lottery $L_1$ (resp. $L_2$) is chosen. The drift component of the motion is related to the difference between expected-like utilities of the lotteries. Time elapsed along the x-axis ("number of sample" denotes "time"), leading to directed paths along it.

process wanders will depend on the specific choice setup. This condition resonates with the fact that, in many situations, utility maximization is computationally intractable [53].

Operationally, we represent choosing between $L_1$ and $L_2$ with a Brownian particle undergoing a continuous random walk [54] that starts at the crossing (taken as the origin) between *4* segments, *2* per lottery, as shown in Fig 2 (to be compared with one segment used in DFT, as shown in Fig 1 along the y-axis, while the x-axis is the time of deliberation). Pictorially, the decision-maker is identified with the Brownian particle itself, whose stochastic path simulates the deliberation act taking place while evaluating the possible alternatives. Each branch encodes information about one lottery outcome—through a potential energy tilting the branch—and its associated probability of occurrence, through the branch length ending with an absorbing boundary. A (perhaps more intuitive) analogous discrete random walk representation is shown in S2 Fig.

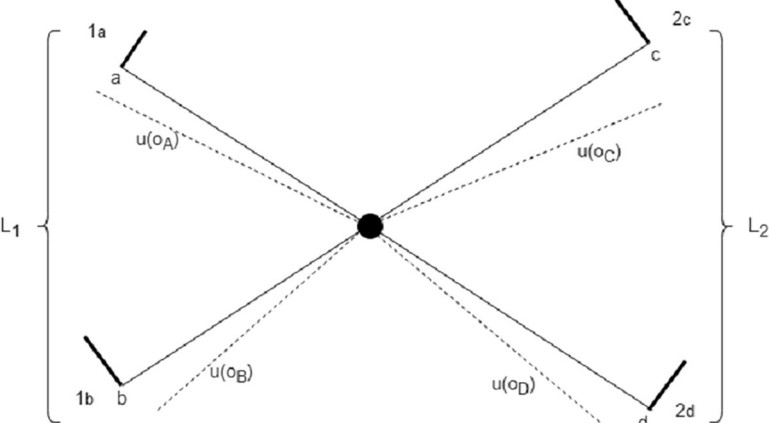

**Fig 2. Stochastic representation of the decision process between lotteries $L_1 = \{o_A, p; o_B, 1-p\}$ and $L_2 = \{o_C, q; o_D, 1-q\}$.** Branches $1_{a/b}$ (resp. $2_{c/d}$) represent the outcomes of $L_1$ (resp. $L_2$) and their related probabilities. The difference between the continuous and dashed lines represent the energy potential associated with the constant forces $\{u(o_A), u(o_B), u(o_C), u(o_D)\}$ exerted on the Brownian particle along each segment $\{1_a, 1_b, 2_c, 2_d\}$ respectively. The thick bars at the end of each branch depict the absorption boundary conditions. The segment lengths $\{a, b, c, d\}$ are determined by the objective probabilities $\{p, 1-p, q, 1-q\}$. The probability of choosing lottery $L_1$ (resp. $L_2$) is given by the probability of being absorbed along branch $1_a$ or $1_b$ (resp. $2_c$ or $2_d$).

When the process is restricted to represent only one lottery, the probability to be absorbed at the end of one branch can be interpreted as the value-distorted subjective probability of the associated outcome (see sub-section "Subjective Probability"). In the presence of two (or more) lotteries, the probability to be absorbed at the end of one branch of a given lottery gives a contribution to the total probability that this lottery is chosen. The probability of choosing lottery $L_1$ (resp. $L_2$)—denoted by $P(L_1)$ (resp. $P(L_2)$)—is thus given by

$$P(L_1) = P(1_a) + P(1_b), \quad P(L_2) = 1 - P(L_1) = P(2_c) + P(2_d) \tag{3}$$

where $P(k_{\eta_k})$ denotes the probability for the particle to be absorbed by the wall located at distance $\eta_k$ on branch $k_{\eta_k}$, for $k = 1,2$ with $\eta_{k=1} \in \{a,b\}$ and $\eta_{k=2} \in \{c,d\}$. In words, the probability of choosing, say, lottery $L_1$ is given by the sum of two terms: the probability of being absorbed along branch $1_a$ – representing $(o_A, p)$ - plus the probability of being absorbed along branch $1_b$, representing $(o_b, p)$.

To quantify the meaning of an outcome $o_A$, we assume the existence of a preference or value function $u(o_A)$, endowed with the minimal standard properties of being non-decreasing and concave on the gain side to represent risk aversion (see sub-section "Risk-seeking behavior for losses" for the loss side). Then, the form of the potential energy acting on the Brownian particle along a branch with outcome $o_A$ is taken as linear, with a slope proportional to $u(o_A)$, as represented by dashed lines in Fig 2. This corresponds to a constant force acting on the Brownian particle along each segment. The sign of the energy potential is such that the greater is an outcome, the higher is the attraction toward the corresponding branch end point.

This representation has the advantage of remaining essentially one-dimensional, the motion on each segment being governed by a simple partial differential equation. For example, the probability density $p(x,t)$ of the particle at position $x$ and time $t$ on branch $1_a$ (of length $a$) evolves according to the following Fokker-Planck equation [55, 56]

$$\begin{cases} \dfrac{\partial p(x,t)}{\partial t} = u(o_A)\dfrac{\partial p(x,t)}{\partial x} + \dfrac{D}{2}\dfrac{\partial^2 p(x,t)}{\partial x^2} \\ p(a,t) = 0 \, \forall t \quad \text{(absorbing boundary)} \\ p(0,t) = f(t) \quad \text{(probability mass from other branches)} \end{cases} \tag{4}$$

where $u(o_A)$ is the constant drift acting on the particle, $D$ is the so-called diffusion coefficient and the two boundary conditions account respectively for the absorbing wall at distance $a$ from the origin,

and $f(t)$ represents the probability of the random walker incoming at the origin from other branches. Note that Eq (4) is just one possible way to look at the problem, i.e. solving a diffusion process on each branch independently and then matching the flux to ensure conservation of probability mass. However, as shown in S1 Appendix, we did not proceed this way: rather, we first solve the absorption problem in the case of only two branches (i.e. a one dimensional Brownian motion between two absorbing walls), and then show how it can be generalized to an arbitrary number of branches.

Simple dimension analysis of Eq (4) shows that $D$ sets the scales for the impact of the outcome values compared with the probabilities in the value formation process: (i) taking very large $D$'s amounts to neglecting the influence of outcome values; (ii) small $D$'s make outcome values dominant in the construction of preferences.

**Explicit expressions for the decision probabilities.** As shown in Fig 2, the probability $P(L_1)$ (resp. $P(L_2)$) for the decision maker to choose lottery $L_1$ (resp. $L_2$) is represented by solving Eq (4) for each of the four branches with the matching condition of the conservation of the probability of presence of the Brownian particle when crossing the junction point at the origin.

Using the theory of random walks and diffusion processes [57], we obtain (see S1 Appendix for derivation)

$$P(L_1) = \frac{\mathcal{U}(L_1)}{\mathcal{U}(L_1) + \mathcal{U}(L_2)}, P(L_2) = 1 - P(L_1) = \frac{\mathcal{U}(L_2)}{\mathcal{U}(L_1) + \mathcal{U}(L_2)} \tag{5}$$

with

$$\mathcal{U}(L_1) = \tilde{\mathcal{U}}_p(o_A) + \tilde{\mathcal{U}}_{1-p}(o_B) = \frac{u(o_A)}{1 - e^{\frac{-2u(o_A)}{pD}}} + \frac{u(o_B)}{1 - e^{\frac{-2u(o_B)}{(1-p)D}}} \tag{6}$$

and

$$\mathcal{U}(L_2) = \tilde{\mathcal{U}}_q(o_C) + \tilde{\mathcal{U}}_{1-q}(o_D) = \frac{u(o_C)}{1 - e^{\frac{-2u(o_C)}{qD}}} + \frac{u(o_D)}{1 - e^{\frac{-2u(o_D)}{(1-q)D}}} \tag{7}$$

Expression (5) recovers the ratio scale representation of Luce's choice axiom for binary choice [27], with effective utilities given by (6) and (7). This implies the so-called strong stochastic transitivity for pairwise choices: $P_{\{L_1,L_2\}}(L_1) \geq .5$ and $P_{\{L_2,L_3\}}(L_2) \geq .5$ imply that $P_{\{L_1,L_3\}}(L_1) \geq max[P_{\{L_1,L_2\}}(L_1), P_{\{L_2,L_3\}}(L_2)]$. Note that the solution of Eq (4) for $N$ alternatives generalizes into

$$P_N(L_j) = \frac{\mathcal{U}(L_j)}{\sum_{i=1}^{N} \mathcal{U}(L_i)} \tag{8}$$

where $P_N(L_j)$ is the probability of choosing lottery $L_j$ among the $N$ available lotteries and the $\mathcal{U}(L_i)$'s are generalized utilities given by expressions of the form (6) and (7). As stated in the introduction, because of the Luce choice structure, our theory would predict a non-deterministic decision when the choice is between two sure outcomes (e.g. $L_1 = \{9,1\}$ vs $L_2 = \{10,1\}$). Therefore, following Luce [27], we assume that no such task is present into the choice set.

As can be seen from (6) and (7), the utility $\mathcal{U}(L)$ of a given lottery is given by the sum of two terms, each representing the attractiveness of an outcome-probability pair, which cannot be decomposed in a simple product of utility and subjective probability, as in expected utility theories. In contrast, probabilities and utilities combine and interact in a non-trivial way, with $D$ quantifying the relative importance of value with respect to probability assigned by the DM. This becomes evident when taking the asymptotic limits of, e.g., $P(L_1)$ (in the presence of another lottery $L_2$ offered as the second option):

$$\lim_{D \to 0} P(L_1) = \frac{u(o_A) + u(o_B)}{u(o_A) + u(o_B) + u(o_C) + u(o_D)}, \text{if} \, u(o_i) > 0 \forall i$$

$$\lim_{D \to \infty} P(L_1) = \frac{\frac{1}{1-p} + \frac{1}{p}}{\frac{1}{1-p} + \frac{1}{p} + \frac{1}{1-q} + \frac{1}{q}} \tag{9}$$

In our framework, a decision maker characterized by $D \to 0$ (resp. $D \to \infty$) is influenced only by outcome values (resp. probabilities), while for finite $D$ his decision derives from an entangled mixture of both. Note that the limit for $D \to 0$ depends on the sign of the utilities: for

example, if $u(o_B)$ and $u(o_D)$ are negative, the asymptotic behavior of the choice probability is

$$P(L_1) \xrightarrow[D \ll 1]{} \frac{u(o_A) + |u(o_B)|e^{\frac{-2|u(o_B)|}{(1-p)D}}}{u(o_A) + |u(o_B)|e^{\frac{-2|u(o_B)|}{(1-p)D}} + u(o_C) + |u(o_D)|e^{\frac{-2|u(o_D)|}{(1-q)D}}} \qquad (10)$$

This shows an intrinsic difference in perception between gains and losses, an asymmetry that we discuss further in sub-section "Probability-distorted effective utility".

## Stage 2: Finite time SRDT

**Rationale.** Many empirical studies (see [58]) have shown how people do not always choose the best option, but the one that gives a fair trade-off between utility and "cost". A decision is in general a stressful operation, and humans have finite computational resources, so even when there is no explicit time constraint for making a choice, low-effort heuristics become attractive as soon as they provide *satisfactory* outcomes. Thus, the time dimension in decision-making cannot be neglected, as static theories of decision-making (including RUM models) do. Next sub-section extends the previously presented model to account for finite time deliberation.

**Theoretical extension.** Eq (5) provided the choice probabilities $P(L_1)$ and $P(L_2)$ for a binary choice between lotteries $L_1$ and $L_2$ assuming infinite available time to make a decision. We are now interested in calculating the choice probability, say $P(L_1)$, **conditioned** on occurring at some time $t \le T$, denoted by $P(L_1|T)$. In other words, $P(L_1|T)$ is the probability to be absorbed by one of the outcomes of $L_1$, **given** that the particle is absorbed somewhere **before** time T. This condition mimics either an explicit time limit (time pressure) or an implicit one, due to accuracy-effort trade-off. Formally, for the binary choice representation in Fig 2, $P(L_1|T)$ is given by

$$P(L_1|T) = P(1_a|T) + P(1_b|T) = \int_0^T dt J_{1a}(a,t) + \int_0^T dt J_{1b}(b,t) \qquad (11)$$

where $J_\eta(x,t)$ is the probability current on branch $\eta$ at position $x$ and time $t$. Given the structure of the problem, a closed form expression of $J_\eta(x,t)$ is hard to obtain. However, a very good approximation of the integrals in (11) is given by the Laplace-transform $\tilde{J}(x,s)$ of the probability current

$$\tilde{J}\left(x, s = \frac{1}{T}\right) = \int_0^{+\infty} dt J(x,t) e^{-\frac{t}{T}} \cong \int_0^T dt J(x,t) \qquad (12)$$

where $s$ is the conjugate variable of time. Therefore, combining Eqs (11) and (12), $P(L_1|T)$ is approximately given by (see S1 Appendix for derivation)

$$P(L_1|T) \cong \frac{\tilde{\mathcal{U}}_p(o_A|T) + \tilde{\mathcal{U}}_{1-p}(o_B|T)}{\tilde{\mathcal{U}}_p(o_A|T) + \tilde{\mathcal{U}}_{1-p}(o_B|T) + \tilde{\mathcal{U}}_q(o_C|T) + \tilde{\mathcal{U}}_{1-q}(o_D|T)}, P(L_2|T) = 1 - P(L_1|T) \quad (13)$$

with

$$\tilde{\mathcal{U}}_p(o|T) = \frac{\sqrt{\left(\frac{u(o)}{D}\right)^2 + \frac{2}{DT}}}{e^{-\frac{u(o)}{Dp}}\sinh\left(\frac{\sqrt{\left(\frac{u(o)}{D}\right)^2 + \frac{2}{DT}}}{p}\right)}$$

(14)

It is easy to check that when there is no time constraint ($T\rightarrow\infty$) Eq (13) retrieves the usual asymptotic choice probabilities in (5).

## Results

### Fourfold pattern of risk preferences

The fourfold pattern of risk preferences [43] is one prominent example of the inadequacy of Expected Utility to describe observed human behaviors. It is experimentally observed that people are: i) risk-averse when gains have moderate probabilities or losses have small probabilities; ii) risk-seeking when losses have moderate probabilities or gains have small probabilities. In Table 1, we report an example of such behavior. Prospect theory, thanks to the interplay of value function and probability weighting, is able to describe it.

To show how our theory can account for such pattern, consider the following decision tasks

$$(A) \; L_1(p) = \{100€, p; 0€; 1-p\} \text{ or } L_2(p) = \{100p€, 1\}$$
$$(B) \; L_3(p) = \{-100€, p; 0€; 1-p\} \text{ or } L_4(p) = \{-100p€, 1\}$$

(15)

Where the probability $p\in[0,1]$. Assume that the decision-maker is characterized by the utility function $u(o) = \frac{1-e^{-ro}}{r}$ for $o\in R$ with constant absolute risk aversion (CARA) $-u''/u'$ equal to the constant $r$, which is everywhere concave, continuous and differentiable. The decision probabilities are given by

$$P(L_1(p)) = \frac{\tilde{\mathcal{U}}_p(100) + \tilde{\mathcal{U}}_{1-p}(0)}{\tilde{\mathcal{U}}_p(100) + \tilde{\mathcal{U}}_{1-p}(0) + \tilde{\mathcal{U}}_1(100p)}$$

$$P(L_3(p)) = \frac{\tilde{\mathcal{U}}_p(-100) + \tilde{\mathcal{U}}_{1-p}(0)}{\tilde{\mathcal{U}}_p(-100) + \tilde{\mathcal{U}}_{1-p}(0) + \tilde{\mathcal{U}}_1(-100p)}$$

(16)

with $P(L_2(p)) = 1-P(L_1(p))$ and $P(L_4(p)) = 1-P(L_3(p))$.

In Fig 3, referring to the example in Eq (15), we show the predicted probability of choosing $L_1$ in task (A) (Fig 3A) and $L_3$ in task (B) (Fig 3B) as a function of $p$, for fixed diffusion coefficient $D$ and different values of $r$. The fourfold pattern is correctly predicted: in Fig 3A, $P(L_1(p))\geq0.5$ for small $p$ (risk-seeking, possibility effect) and $P(L_1(p))\geq0.5$ for large $p$ (risk-averse, certainty effect). The situation is reversed in Fig 3B. Note that, despite the fact that our model is structurally different from Expected Utility, a more concave utility function, i.e.

**Table 1. Example of the fourfold pattern of risk attitudes.**

|  | Gains | Losses |
|---|---|---|
| High probability (certainty effect) | $\{100€,0.95;0,0.05\}\prec\{95,1\}$ | $\{-100€,0.95;0,0.05\}\succ\{-95,1\}$ |
|  | Risk-Averse | Risk-Seeking |
| Low probability (possibility effect) | $\{100€,0.05;0,0.95\}\succ\{5,1\}$ | $\{-100€,0.05;0,0.95\}\prec\{-5,1\}$ |
|  | Risk-Seeking | Risk-Averse |

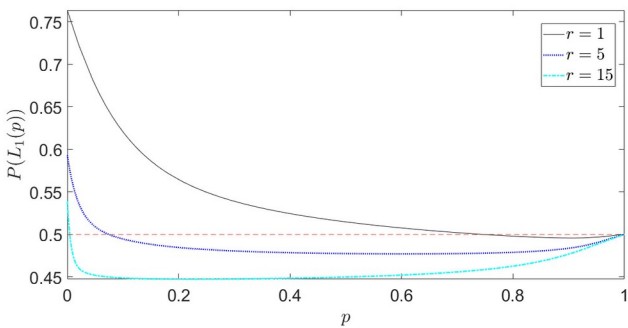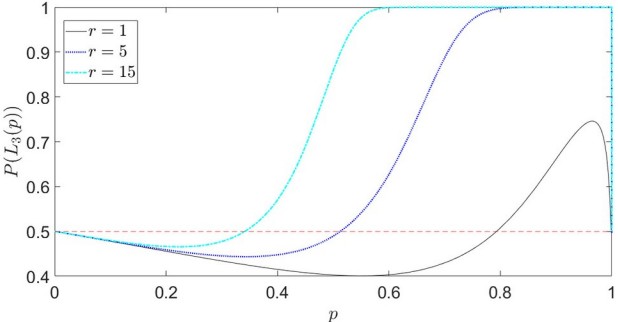

**Fig 3. Fourfold pattern of risk attitudes: Choice probabilities as a function of outcome probabilities (Eq (15)).** (a) $P(L_1(p))$ for task (A); (b) $P(L_3(p))$ for task (B). In this example we set $D = 10$ and scan different values of the CARA coefficient $r$.

higher $r$, leads to greater risk-aversion in the gain domain. However, $r$ is not an absolute indicator of risk-aversion as in EU, since the ultimate choice probabilities will depend also on the values of $D$ and $T$ (when time constraints are considered). Interestingly, our model predicts also that greater risk-aversion for gains corresponds to greater risk-loving for losses, suggesting a positive correlation (some evidence for such correlation is reported in [59]).

By further inspection of Fig 3A, we see that something weird happens: the choice probability $P(L_1(p))$ does not go to 0.5 as $p \to 0$. But this should actually be expected, because $L_1$ and $L_2$ in Eq (15) become identical in this limit. This is due to the fact that the contribution to the choice probability from outcome 100, $\tilde{\mathcal{U}}_p(100)$, does not go to 0 as p goes to 0. Thus, there is still a probability to be absorbed along that branch. Conversely, when $p$ is exactly 0, as in $L_2$, there is no branch corresponding to such outcome. As next subsection will explain, this amounts to an infinite overweighting of small probabilities. Technically, this problem is known as a singular perturbation limit [60], where, informally, "the solutions of the problem at a limiting value of the parameter are different in character from the limit of the solutions of the general problem" [61]. In this case, the singular perturbation is characterised by the following inequality

$$\lim_{p \to 0} \tilde{\mathcal{U}}_p(o) = u(o) \neq \tilde{\mathcal{U}}_0(o) (\text{if } u(o) > 0) \qquad (17)$$

Such singularity is removed once we include finite time constraints, i.e. by imposing that the decision cannot take an infinite time. Indeed, replacing the effective utilities in Eq (6) with the time-constrained ones in Eq (14), the contribution to the choice probability of the probability $p$ outcome satisfies the following limit

$$\lim_{p \to 0} \tilde{\mathcal{U}}_p(o|T) = 0 \text{ if } T < \infty \qquad (18)$$

Eq (18) means that, when the probability of an outcome goes to 0, the corresponding probability to be absorbed along that branch also goes to 0, not contributing to the choice probability of the related lottery. Let us stress that we do not impose a "small" value of $T$ to get rid of the singularity; $T$ can be arbitrarily large, but finite.

The probability of choosing $L_1(p)$ in Eq (15), given that the decision occurs before $T < \infty$, reads

$$P(L_1(p)|T) = \frac{\tilde{\mathcal{U}}_p(100|T) + \tilde{\mathcal{U}}_{1-p}(0|T)}{\tilde{\mathcal{U}}_p(100|T) + \tilde{\mathcal{U}}_{1-p}(0|T) + \tilde{\mathcal{U}}_1(100p|T)} \qquad (19)$$

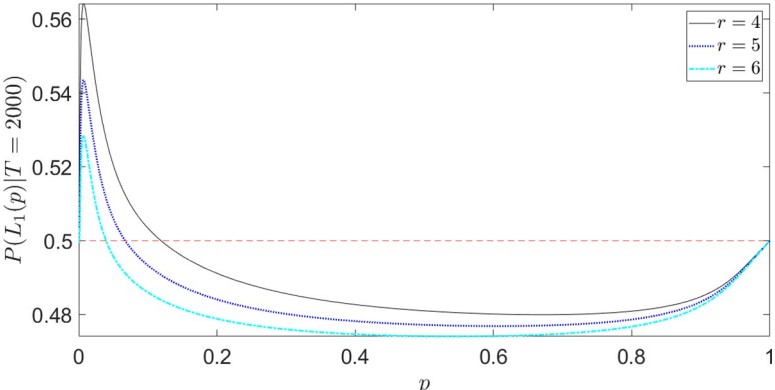

**Fig 4. Fourfold pattern of risk attitudes (gain-side): Time-constrained choice probabilities as a function of outcome probabilities (Eq (19)).** Differently from Fig 3A (first line of Eq (16)), $P(L_1(p)|T) \to 0.5$ *as* $p \to 0$, as it should. In this example we set $D = 10$, $T = 2000$ and scan different values of the CARA coefficient $r$.

Fig 4 shows the choice probability in Eq (19) for different values of $r$ and fixed $T$. The gain-side of the fourfold pattern of risk preferences is now correctly described; differently from Fig 3A (first line of Eq (16)), $P(L_1(p)|T) \to 0.5$ *as* $p \to 0$, as it should.

Let us now investigate the role of the other parameters, the diffusion coefficient $D$ and the time constraint $T$. As said in Section "Explicit expressions for the decision probabilities", $D$ is a kind of "utility-numeraire", determining the relative impact of the outcome values compared with the probabilities in the value formation process. The role of $T$ is more subtle: as we will see in the next subsection, a smaller $T$ implies more underweighting (resp. overweighting) of small (resp. high) outcome probabilities. Figs 5 and 6 show the choice probabilities in Eq (16) for different values of $D$ and $T$, respectively. On the gain-side (Fig 5A), as $D$ decreases, the strength of preferences increases and the preference reversal point between risky and safe lottery shifts to the right (risk-seeking for a wider range of $p$'s). On the loss-side (Fig 5B), decreasing $D$ shifts the curve upward and leftward, implying stronger risk-seeking preferences for a wider range of $p$'s.

Focusing now on Fig 6, we see that, through the underweighting (resp. overweighting) of small (resp. high) probabilities, a smaller $T$ destroys both the possibility effect on the gain-side (no risk-seeking behavior for low $p$) and the certainty effect on the loss-side (no risk-seeking behavior for high $p$). In general, a smaller $T$ implies greater risk-aversion, as we will discuss in subsection "Preference reversal with time pressure".

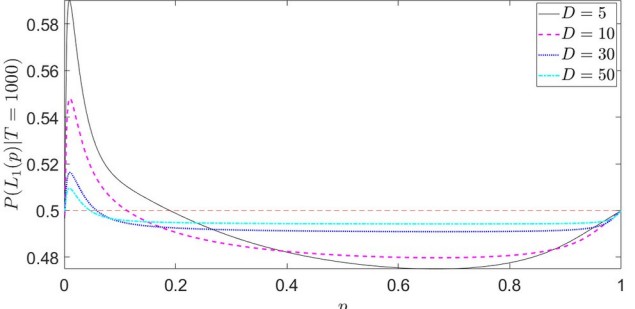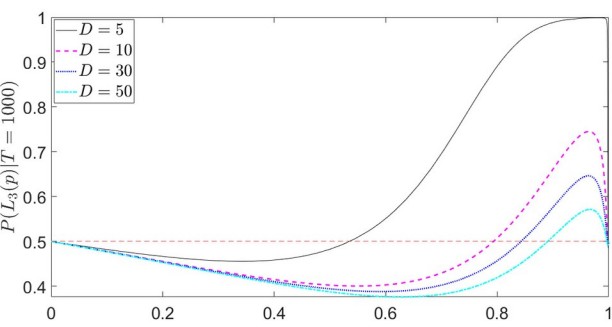

**Fig 5. Fourfold pattern of risk attitudes: Time-constrained choice probabilities for different values of $D$ (Eq (19)).** (a) $P(L_1(p)|T)$ for task (A); (b) $P(L_3(p)|T)$ for task (B). In this example we set $r = 4$, $T = 1000$.

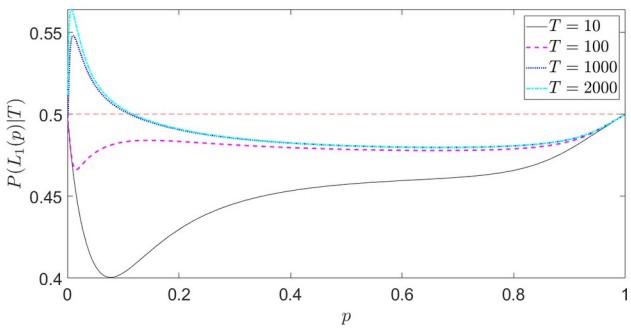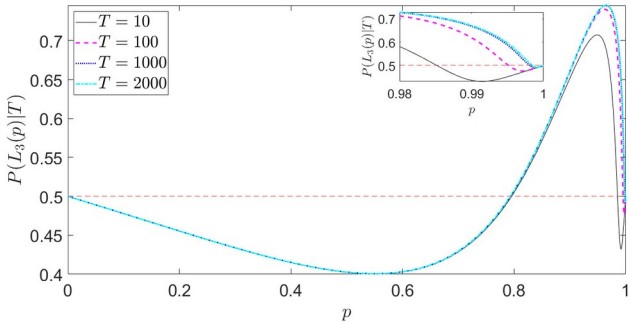

**Fig 6. Fourfold pattern of risk attitudes: Time-constrained choice probabilities for different values of $T$ (Eq (19)).** (a) $P(L_1(p)|T)$ for task (A); (b) $P(L_3(p)|T)$ for task (B) (the inset shows the choice probabilities for high-values of $p$). In this example we set $r = 4$, $D = 10$.

As stated at the beginning of the Section, in prospect theory, the fourfold pattern of risk preferences is usually explained in terms of the combined effects of probability weighting and a convex-concave value function. The next three subsections show how, although in our model these two constructs are not separable, it is still possible to identify them as consequences rather than postulates, offering an additional intuition on why our theory can explain such patterns. Specifically, the study of how value and time constraints affect probability perception is discussed in subsections "Subjective Probability without time-contraints" and "Realistic Inverse double-S-shaped probability weighting function". Conversely, the effect of probability on value perception is analyzed in subsection "Probability-distorted effective utility".

**Subjective probability without time-contraints.** Eq (5), together with (6) and (7), show the resulting form of the decision probabilities without time-constraints for the binary risky choice (1). From here, we now focus on studying the predicted probability perception of the Decision-maker (DM), say, of outcome $o_A$ of lottery $L_1$. A convenient way to extract such information is to look at the probability of absorption along branch $1_a$, conditional on being absorbed along any branch pertaining to $L_1$

$$\pi(p) := P(1_a|L_1) = \frac{\tilde{\mathcal{U}}_p(o_A)}{\tilde{\mathcal{U}}_p(o_A) + \tilde{\mathcal{U}}_{1-p}(o_B)}$$

$$\tilde{\mathcal{U}}_p(o_A) := \frac{u(o_A)}{1 - e^{\frac{-2u(o_A)}{pD}}}, \quad \tilde{\mathcal{U}}_{1-p}(o_B) := \frac{u(o_B)}{1 - e^{\frac{-2u(o_B)}{(1-p)D}}}$$

(20)

$\pi(p)$ can be interpreted as the amount of attention devoted to outcome $o_A$ when the DM is looking at lottery $L_1$. Several authors [62, 63] have established connections between subjective probability and similar psychological notions. Indeed, the fact that $\pi(p)$ defined in (20) represents a meaningful measure of subjective probability is supported by its asymptotic limits as a function of $D$:

$$\lim_{D \to \infty} \pi(p) = p$$

(21)

$$\lim_{D \to 0} \pi(p) = \frac{u(o_A)}{u(o_A) + u(o_B)}, \quad \text{if } u(o_A), u(o_B) > 0$$

For $D\to\infty$, outcome values (potential energies) become negligible compared with the stochastic component and the probability perception is unaltered, so that the subjective probability is equal to the objective one. In contrast, for $D\to0$, the decision maker does not pay attention to the probabilities and focuses solely on the payoffs, interpreting their likelihood only as a function of their magnitude. As for Eq (9), a simple interpretation of the $D\to0$ limit is possible only when both utilities are positive. For $u(o_A)$ and $u(o_B)$ negative, the expression becomes

$$\pi(p) \underset{D\ll1}{\to} \frac{|u(o_A)|e^{\frac{-2|u(o_A)|}{pD}}}{|u(o_A)|e^{\frac{-2|u(o_A)|}{pD}} + |u(o_B)|e^{\frac{-2|u(o_B)|}{(1-p)D}}} \tag{22}$$

Eq (22) implies that when negative utilities are involved, the decision maker, even in the $D\to0$ limit, takes into account the event probabilities.

For finite non-zero $D$, an interesting value-distortion of probability perception arises: Fig 7 shows $\pi(p)$ vs $p$ for different $\frac{u(o_B)}{u(o_A)}$ and $D$ values. Our theory thus derives the empirical inverse S-shape of subjective probability as a function of objective probability, for instance used in standard Prospect Theory by Tversky and Kahneman [11], indicating that human beings tend to overestimate rare events and underestimate high probability events. More specifically, $\pi(p)\geq p$ (resp. $\pi(p)<p$ for $p\leq p^*$ (resp. $p>p^*$), where $p^*$ is the inflection point of $\pi(p)$ given by $p^* = \frac{u(o_A)}{u(o_A)+u(o_B)}\left(\frac{\partial^2\pi(p)}{\partial p^2}\big|_{p=p^*} = 0\right)$. Our theory predicts that the asymmetry in the distortion of $\pi(p)$ for $p\to0$ and $p\to1$ is controlled by $\frac{u(o_A)}{u(o_B)}$: the larger this ratio is, the larger is the subjective distortion for small $p$'s compared with large $p$'s.

There is empirical evidence that changing lottery payoffs changes inflection points. In [64], for each individual, the authors perform the elicitation of two probability weighting functions $\pi_S^-(p)$ and $\pi_L^-(p)$ for gambles involving small and large losses. The idea is that, when considering lotteries like

$$\begin{aligned} L &= \{-1000\text{€}, 0.1; -10\text{€}, 0.9\} \\ L' &= \{-1000000\text{€}, 0.1; -10000\text{€}, 0.9\} \end{aligned} \tag{23}$$

it is possible that the probability 0.1 is not weighted in the same way, because of the different magnitude of the consequences and because of the "distance" between the lottery outcomes. Note that Rank-dependent models (e.g. CPT) predict that $\pi(0.1)$ is the same in both lotteries,

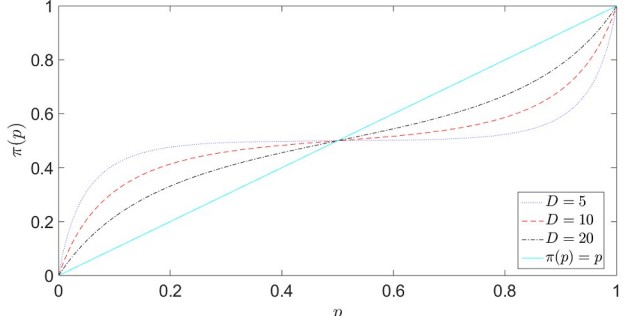
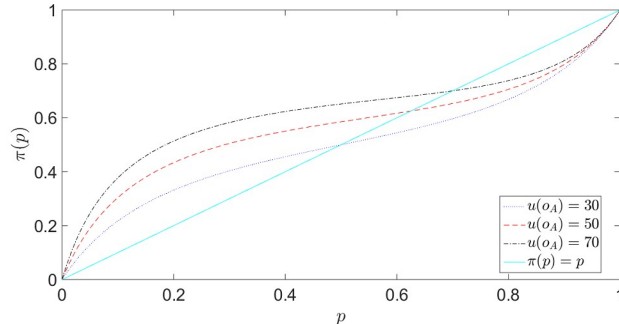

**Fig 7. Value-distorted subjective probability $\pi(p)$ given by expression (20): (Left) fixed u(o_A) = u(o_B) = 30, D is varied.** As $D$ grows, $\pi(p)\to p$; (right) fixed $D = 20$ and $u(o_B) = 30$, $u(o_A)$ is varied. As $u(o_A)$ grows, the inflection point $p^* = \frac{u(o_A)}{(u(o_A)+u(o_B))}$ shifts toward the right while the curve shifts upward. Varying $u(o_B)$ gives symmetrical behaviors.

since $L$ and $L'$ are comonotonic. On average, the authors find that small probabilities ($\leq 0.33$ for small losses and $\leq 0.5$ for large losses) are overweighted (indicating pessimism). The usual inverse-S shape thus holds over both small and large losses, but the inflection point shifts to the right over large losses.

While deriving or recovering the empirical inverse S-shape, our formulation of the subjective probability is fundamentally different from those used in existing decision theories, such as the Prelec II weighting function [65] parametrized to account for some assumed probability distortion, which is supposed to be intrinsic to the DM and can be determined by calibration of the results of a number of standard tests and questions presented to the DM [66]. These subjective probabilities are considered independent of the values of the outcomes to which the probabilities are associated. We have previously argued and also referred to empirical evidence that there is no such thing as an outcome without value. Even a question as far from every life on the probability of life on Mars, say, carries, depending on the DM, religious, scientific, and cultural values and possibly more. In our framework, the subjective probability (20) is influenced by the outcome values and represent the contribution of each outcome in a lottery to the choice of that lottery by the DM. Thus, our theory suggests that it is ill-conceived to attempt characterizing the subjective probabilities of DM. Our approach allows us to formulate a general hypothesis that subjective probabilities are value-dependent, which deserve empirical investigations. In existing decision theories, the subjective probabilities are multiplied by the utilities of the associated events to form a measure of worth and then the choice probability "layer" is added on top. In our theory, subjective probabilities are instead encapsulated into decision probabilities, the former determining the latter. Our model can thus be viewed as a natural generalisation beyond the standard factorisation of probabilities and values to form value preferences.

At this stage, the definition of subjective probability as a relative absorption probability (Eq (20)) may seem somewhat counterintuitive, notwithstanding the fact that it correctly retrieves the objective event probability in the $D \to \infty$ limit. We would like to stress that our mathematical formulation of the subjective probability is fundamentally different from that in expected utility theories. In Expected utility, as described by Savage [67], the assumption of separation between preferences and beliefs is crucial for the elicitation of subjective probability. However, as stated in the introduction, the simultaneous estimation of utility and subjective probability is subjected to the joint hypothesis-testing problem [20], and many methods have been devised to circumvent such issues [68, 69]. Here, in contrast, the subjective estimation of the likelihood of an event depends on the associated magnitude. Consequently, in our model, the subjective probability is actually *implied* by the utility function, and thus two separate functions cannot be really identified. Our definition of subjective probability should be treated as a way to extract how the choice probability depends on the outcome probabilities, and to get an intuition on why our model is able to explain the fourfold pattern of risk preferences. Concretely, one would just need to estimate the utility function (together with the parameters D and T), and the corresponding "belief function" comes as a result. The next subsection presents an analysis of the subjective probability when time or "energy" constraints are considered.

**"Realistic" inverse double-S-shaped probability weighting function.** Al-Nowaihi and Dhami [48] report that a theory of choice should be able to describe the following two stylized facts: i) overweighting low probability events and underweighting high probability ones; ii) neglecting extremely low probability events and considering as certain extremely probable events. The first fact is essentially captured by an inverse S-shaped probability weighting function, as derived in Eq (20). The second one is referred by Kahneman and Tversky [43] as an *editing phase*: "the simplification of prospects can lead the individual to discard events of extremely low probability and to treat events of extremely high probability as if they were

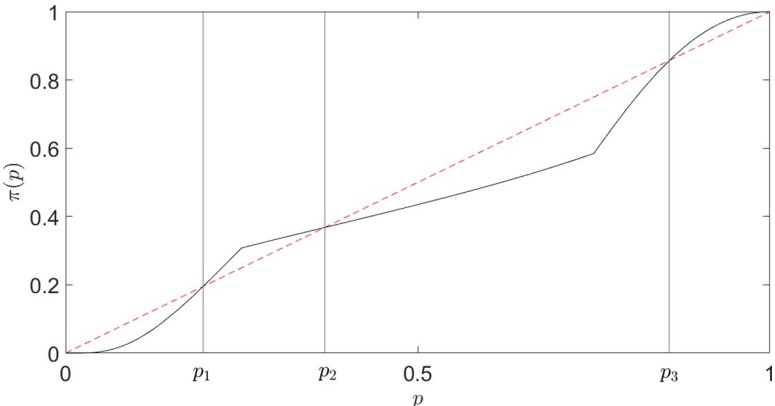

**Fig 8. An example of a composite Prelec function postulated in [48].** The DM underweights extremely low probabilities $[0,p_1]$ and overweights extremely high probabilities $[p_3,1]$. For $p \in [p_1,p_3]$, the DM overweights low probabilities $[p_1,p_2]$ and underweights high probabilities $[p_2,p_3]$.

certain". Clearly, this resonates with the idea that the DM has limited computational resources and, even when there is no explicit time limit, the processing cost acts as such.

To account for both patterns i) and ii), Al-Nowaihi and Dhami axiomatically construct a *composite* probability weighting function, shown in Fig 8, obtained by the concatenation of three different Prelec functions, for a total of 6 parameters (see Eq 6.2 in [48]). A DM with such probability function underweights (ignores) very low probabilities events – $p \in [0,p_1]$ – and overweights (considers as certain) extremely probable events - $p \in [p_3,1]$ –reflecting stylized fact ii). Within the middle range $p \in [p_1,p_3]$, the function has an inverse-S shape, addressing stylized fact i).

Although the proposed probability weighting function addresses the previously mentioned stylized facts, it has six parameters and may seem ad-hoc and artificial. Our framework, on the other hand, *predicts* the desired shape, resulting from the superposition of two effects: finite-time deliberation and value distortion. Indeed, referring to the previously derived value-distorted subjective probability (20) for outcome $o_A$ of lottery $L_1$ in (1), the time-dependent generalization is (approximately) given by

$$\pi(p|T) \cong \frac{\tilde{\mathcal{U}}_p(o_A|T)}{\tilde{\mathcal{U}}_{1-p}(o_B|T) + \tilde{\mathcal{U}}_p(o_A|T)} \tag{24}$$

with $\tilde{\mathcal{U}}_p(o|T)$ given in Eq (14). In Fig 9, we plot $\pi(p|T)$ for different values of $T$, fixing $u(o_A) = u(o_B) = D = 10$ for illustrative purpose. We can see how the value-distortion and the finite time deliberation play opposite effects: for high values of $T$, one can observe an inverse S-shape, due to the influence of value on probability perception (as in Fig 7). For low values of $T$, the influence of time pressure becomes dominant, resulting into a S-shaped probability weighting. For intermediate values of $T$ (in the example $T = 0.3$, black star-dotted line), the superposition of these two "forces" results in an inverse double S-shaped probability weighting, similar to the one in Fig 8.

In summary, our framework predicts the probability weighting function postulated in [48] with only 2 parameters–$D$ and $T$- and offers a more microscopic explanation for such observed behavior, in terms of competition between value-distortion and finiteness of computational resources. Let us stress that the time constraint in our model is not necessarily meant as an external time pressure, but it can also be conceived as an internal time pressure, because of

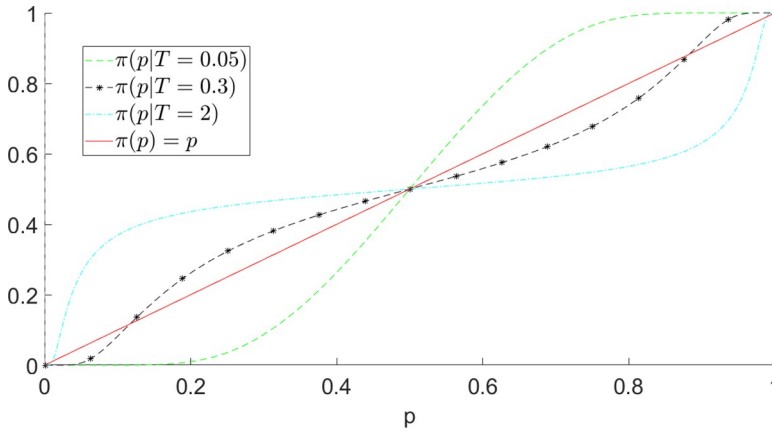

**Fig 9. Predicted time-dependent subjective probability (20) for different values of T ($o_A = o_B = D = 10$).** For high (resp. low) values of $T$, the value-distortion (resp. finite time deliberation) effect dominates. For intermediate values of $T$, an inverse double S-shaped probability function is predicted, similar to the one postulated in [48].

energy constraints and efficiency-accuracy tradeoff. Therefore, at this stage, we are not claiming that an explicit time-pressure is needed to recover an inverse double-S curve.

**"Probability-distorted" effective utility.** In the previous sub-section, we have studied how the outcome values alter the probability perception of the DM. Here, we study the effect of probability on value perception. Eq (5) has introduced the effective utilities $\tilde{\mathcal{U}}_p(.)$, which are transformed from the utilities $u(.)$ via a non-trivial nonlinear operation involving the outcome probabilities. This corresponds to the dual of the value-distorted probability $\pi(p)$ given in expression (20) in the form of a value perception $\tilde{\mathcal{U}}_p(.)$ influenced by probability. Fig 10 shows the transformed utility function $\tilde{\mathcal{U}}_p(.)$ as a function of the original one $u(.)$ for different values of $D_p := \frac{pD}{2}$.

The interaction between probabilities and values transforms an initially risk-averse (concave) utility function $u(.)$ into a convex risk-seeking utility on a sub-interval of the loss domain, *predicting* the existence of a reference point to discriminate between behavior toward

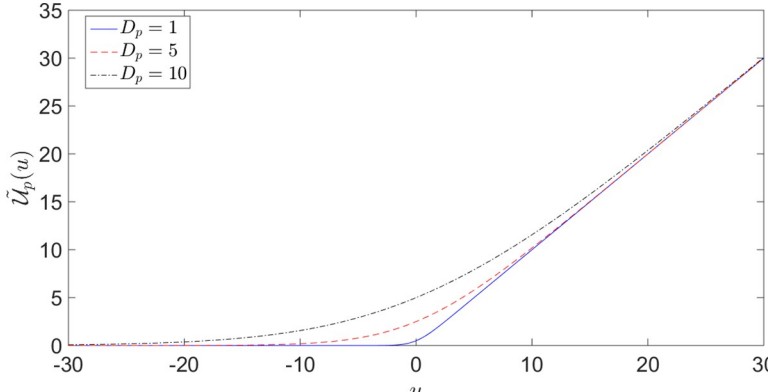

**Fig 10. Transformed utility function $\tilde{U}_p(.)$ (20) as a function of the original one $u(.)$ for different values of $D_p := \frac{pD}{2}$.** The distortion is significant for negative $u$ and for small positive $u$ (i.e. small $u$ compared to $D_p$) and is all the stronger, the smaller is $|u/_{D_p}|$. The transformed utility flattens out for large negative $u$. $\tilde{\mathcal{U}}_p(.)$ approaches $u$ asymptotically for large positive values ($\tilde{\mathcal{U}}_p(.) \to u$).

gains and behavior toward losses, as postulated in prospect theory. We stress here that this comes as another prediction of the theory without any parameter adjustment or added ingredients. In particular, it is not a phenomenological assumption put in the theory, for instance as in Prospect Theory.

To illustrate this effect quantitatively, let us consider again the utility function $u(o) = \frac{1-e^{-ro}}{r}$ for $o \in R$. The corresponding transformed utility function $\tilde{\mathcal{U}}_p(o)$ given by expression (20) reads

$$\tilde{\mathcal{U}}_p(o) = \frac{(1 - e^{-ro})/r}{1 - \exp\left\{-\frac{1}{rD_p}\left(1 - e^{-ro}\right)\right\}}, \text{ with } D_p := \frac{pD}{2} \tag{25}$$

Fig 11 shows $\tilde{\mathcal{U}}_p(o)$ of Eq (25) for different values of $D_p(r = 1)$. The presence of an inflection point $o_r^*(D_p)$ implies a risk-averse concave portion for $o > o_r^*(D_p)$ and a convex risk-taking behavior for $o < o_r^*(D_p)$ (in particular for losses). Therefore, transformation (20) and (25) predict a risk taking behavior on the loss side even when starting with a utility function that is everywhere concave, in agreement with the outlined predictions on the fourfold pattern of risk-preferences [43].

Decision tasks like those in Eq (15) are classic examples where the weak risk-aversion relation, denoted by $R_w$, can be applied:

$$L_2 R_w L_1 \Leftrightarrow E[L_2] = E[L_1] \text{ and } L_1 \text{is a degenerate lottery} \tag{26}$$

meaning that $L_2$ is riskier than $L_1$. More general relations [70] have been suggested to formalize risk-aversion, such as the so-called strong risk-aversion (or second-order stochastic dominance) $R_s$:

$$L R_s L' \Leftrightarrow L = L' + \epsilon \text{ where } \epsilon \text{is a white noise} \tag{27}$$

Within expected utility, these two definitions of risk-aversion coincide [70, 71], but, in general, when departing from the expectation structure, the two relations differ [72] and need to be studied separately. An example for strong risk-aversion is the following:

$$L_5 = \{0\euro, 0.75; 2\euro; 0.25\} \text{ or } L_6 = \{0\euro, 0.5; 1\euro; 0.5\} \tag{28}$$

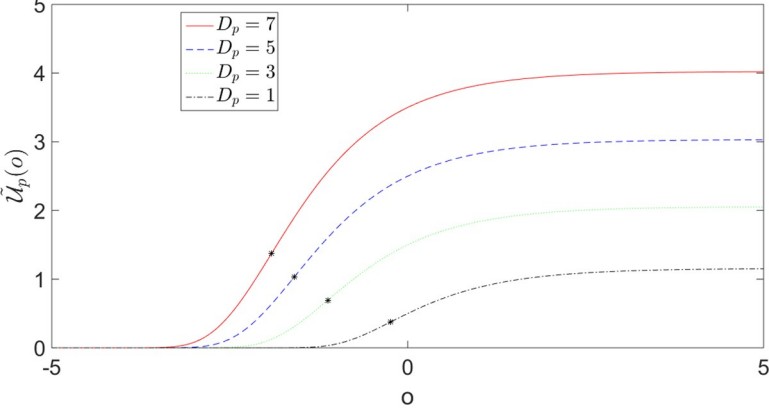

**Fig 11. Transformed utility $\tilde{U}_p(o)$ (25) for different values of $D_p := \frac{pD}{2}$ ($r = 1$).** Black star points indicate the location of the inflection points $o_r^*(D_p)$, where the function changes the sign of its concavity.

According to (27), $L_5$ is riskier than $L_6$; our framework, using for example $r = 1$ as before, predicts the correct pattern for the majority of $D$'s:

$$P(L_6) \geq P(L_5) \forall D \geq 1 \tag{29}$$

In summary, without added assumptions, our theory predicts what has been postulated for instance by prospect theory, with a concave part of the value function for gains and a convex part for losses. These properties derive naturally from the stochastic representation of probabilities in the presence of values.

However, the way in which the above-presented transformed utility determines choice preferences is different from usual decision models; in expected utility, the concavity of utility function implies risk aversion through Jensen's inequality [73]. Here, due to the non-linear form of $\hat{\mathcal{U}}_p(o)$, it is in general not easy to derive analogous simple constraints for the model parameters.

More generally, we stick with the notion of an (initially concave) utility for the following reason: utility is a well-defined concept in choice under certainty [74], where diminishing marginal utility indicates less and less increase in "happiness" as wealth increases. In Expected Utility, the concept of diminishing marginal utility and risk-aversion are fundamentally entangled: there cannot be one without the other. On the other hand, in generalized theories like Rank-Dependent Utility Theory, as shown in [75], it is possible for a decision-maker to be risk-seeking with a concave utility function, provided the probability weighting is sufficiently "optimistic". Therefore, the concept of diminishing marginal utility and risk-aversion are decoupled to same extent. Analogously, our model hypothesis is that the utility function, when in the context of choice under certainty, has some form (e.g. the CARA function used in the manuscript), expressing (or not) diminishing marginal utility. Then, as soon as there is some uncertainty, due to the interaction between probability and value, the utility becomes "distorted", and assumes a form like the one in Eq (25), allowing to exhibit risk-seeking behavior for losses.

## Stochastic dominance

First order Stochastic Dominance [76] is a property that decision theorists usually are not willing to give up, as it essentially encodes the reasonable behaviour that "more is better". A random variable (gamble) $L_1$ has first-order stochastic dominance over gamble $L_2$ if $P(L_1 \geq o) \geq P(L_2 \geq o) \forall o$ and for some $o$ $P(L_1 \geq o) > P(L_2 \geq o)$, where $\{o\}$ is the set of possible outcomes. However, people often violate it when presented with choices like

$$L_1 = \{96€, 0.90; 14€; 0.05; 12€; 0.05\} \text{ or } L_2 = \{96€, 0.85; 90€; 0.05; 12€; 0.10\} \tag{30}$$

Even if $L_1$ stochastically dominates $L_2$, most people choose $L_2$. Popular decision models like rank-dependent utility theory [10] and cumulative prospect theory [11] cannot account for this pattern. Within our framework, this is explained when DM exhibit relatively low values of $D$, such that the decision is "value-oriented" and the DM does not pay sufficient attention to the probabilities. For this particular gamble, assuming linear utility function, $P(L_2) \approx 0.65$ for small values of $D$, quite close to the fraction *70%* of people choosing $L_2$ experimentally found by Birnbaum and Navarrete [77]. Note that our model does not predict any violation when the dominance is "evident'", as in the following examples

$$\begin{aligned}
&\text{(A) } L_1 = \{1€, 0.5; 3€; 0.5\} \text{ or } L_2 = \{1€, 0.5; 2€; 0.5\} \rightarrow P(L_1) \geq P(L_2) \forall D \\
&\text{(B) } L_3 = \{11€, 0.5; 12€; 0.5\} \text{ or } L_4 = \{10€, 1; 0€; 0\} \rightarrow P(L_3) \geq P(L_4) \forall D
\end{aligned} \tag{31}$$

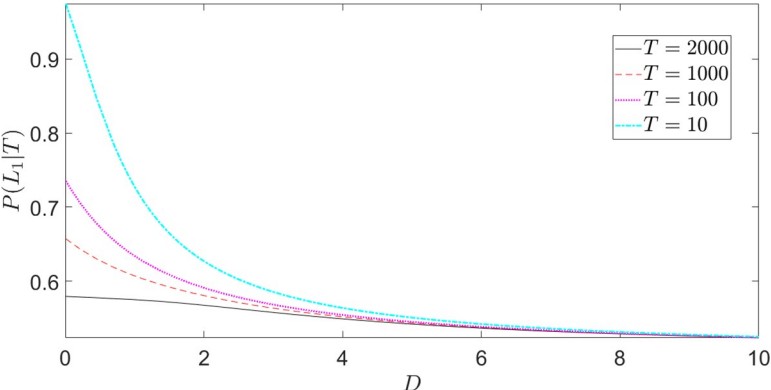

**Fig 12. Probability to choose the dominant option $L_1$ in Eq (31).** In the example, a linear utility function was
assumed. For smaller values of both $D$ and $T$, the choice becomes more deterministic.

It is clear that $L_1$ dominates $L_2$ in (A) and $L_3$ dominates $L_4$ in (B), and people choose accord-
ingly. Fig 12 shows the predicted probability to choose $L_1$ in task (A) (Eq (31)) for different val-
ues of $D$ and $T$. As expected, for small values of the diffusion coefficient $D$, the choice becomes
more deterministic. The (explicit or implicit) time constraint $T$ plays a similar role.

Several descriptive theories [78, 79] allowing violations in cases like (30) predicted unrea-
sonable behaviour in tasks like (31), essentially because two outcomes with the same objective
probability were forced to have the same subjective one [80]. Our framework, thanks to the
non-separable form of the lotteries' attractiveness, avoids this problem and confirms its signifi-
cant predictive power.

## Predictions from finite time SRDT

This Section presents further predictions of our theory when generalized to account for finite
decision time.

**Inverse relation between choice probability and response time.** Several studies (e.g.
[45]) report that there is an inverse relation between the probability to choose an option and
the (mean) decision time to choose that option (see Fig 13). Intuitively, the more "difficult" the
choice (e.g. two lotteries with similar expected values), the more time it will take to decide, and

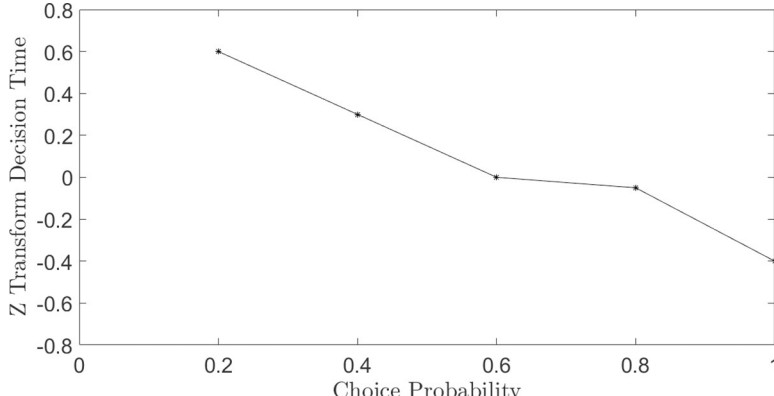

**Fig 13. Mean time to choose an action as a function of the probability to choose that option.** A standardized (z-
score) scale is used for response times, as in [35]. The data are from [45].

the choice probability will be around .5, because the optimal decision is not obvious. By construction, our theory predicts such phenomenon, since:

$$E[T|ChooseL_1] = \frac{E[T]}{P_{\{L_1,L_2\}}(L_1)} \tag{32}$$

where $E[T]$ is the mean decision time to choose any option and $E[T|ChooseL_1]$ is the mean time to choose option $L_1$.

**Preference reversal with time pressure.** Experimental studies [46, 47, 81–83] investigating decision making under time pressure have shown that choice probabilities change as a function of the time limit imposed by the experimenter. Specifically, the probability of choosing an option can shift from below .50 to above .50 (or vice versa) depending on time pressure.

In [46], the main result is that the fraction of subjects choosing low risk gambles increased from below .50 (low time pressure) to above .50 (high time pressure). Therefore, subjects essentially became more risk-averse as the time available to make a decision decreases. Our framework is able to predict such pattern. Consider a choice of the form:

$$L_1 = \{180€, \ 0.5; \ 20€, \ 0.5\} \ vs \ L_2 = \{180€, \ 0.5; \ 30€, \ 0.25; \ 15€, \ 0.25\} \tag{33}$$

where $E[L_1]<E[L_2]$, but $Var(L_1)<Var(L_2)$, so $L_2$ gives on average a greater payoff, but is riskier. Assume for simplicity a linear utility function $u(x) = x$. In Fig 14 we can see clearly that $P(L_1|T)$ goes from below .5 to above .5 as time pressure increases (i.e. time available decreases), reflecting the tendency found in [46] of increasing risk-aversion as a function of time pressure. Note that while Decision Field Theory needs to assume an asymmetric starting point for the random walk in order to capture a preference reversal, our theory essentially predicts this pattern without adjusted additional parameters.

## Discussion

We have presented a simple and efficient "stochastic representation" framework that describes the human decision-making process as inherently probabilistic. It is based on a representation of the deliberation process leading to a choice through stochastic processes, the simplest of which is a random walk. Differently from random utility theory (external noise added to the rational utility and probability representation as a calibration procedure), our stochastic representation framework relies on a plausible description of the (assumed) intrinsic stochasticity of

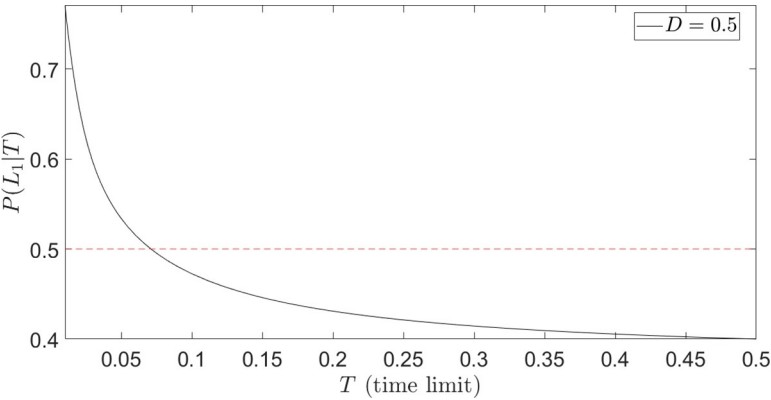

**Fig 14. Probability to choose the low-risk gamble $L_1$ as a function of time pressure.** The shorter the available time (i.e. the higher the time pressure), the higher the probability to choose the low-risk gamble. The time units here are arbitrary and just illustrative, since they depend on the diffusion coefficient $D$.

the human choice process. Our proposed approach does not disentangle probability and value as in expected utility theories, rather it allows interaction between them in a non-trivial way. Despite its simplicity, the model provides straightforward derivations at a more microscopic level of several known structures that have been documented empirically in human decision theory. Our theory also provide a number of novel predictions.

Here, only structural properties have been presented through simple examples, which are not sufficient to falsify the theory. At this stage, the parsimony of its formulation and the wealth of obtained properties, which are in qualitative or semi-quantitative agreement with empirically observations, makes our theory interesting to further explore. We plan to use more sophisticated procedures to test our model against the major decision theories, based on cross-validation methods: parameters are first estimated from one part of an experiment, and then these same parameters are applied to a separate part of the experiment and the predictions are evaluated. Note that we cannot use the usual Wilks likelihood-ratio test [84] because in general the models will not be nested, but other methods are possible, such as the Vuong test [85] and Information Criteria (AIC [86] and BIC [87]).

The current formulation is not meant to be "the" definitive framework (if it would exist)—since as already mentioned it presents some limitations, such as those deriving from Luce choice axiom—but a baseline to construct more elaborate models, keeping in mind the trade-off between parsimony and explanatory power.

In general, we are aware that testing alternative ways of value formation is very difficult, because of the measurement problem in economics [88]. Indeed, we cannot really measure the "degree of happiness" of the decision-maker, but we have to infer it–adopting one particular model—through her choices. This adds an additional layer of complexity with respect to other hard sciences, such as physics or chemistry. On the other hand, contribution of this type may help to devise more effective ways to elicit preferences, deepening our understanding of decision processes. In addition, further theoretical and empirical work may lead to modifications of the presented theory, where the expected utility hypothesis (separability of probability and value) can be seen as a particular case of a more complex structure, where probability and value do interact to some extent in the decision maker's mind.

## Supporting information

**S1 Fig. Discrete random walk analogue of a one-dimensional drifted Brownian motion in presence of two absorbing boundaries.**
(TIF)

**S2 Fig. Random walk analogue of the continuous stochastic representation of choice between two binary lotteries.** The transition probabilities are different for each segment, in order to correctly represent the different outcome-dependent potentials, while the distance of the absorbing states from the centre is different in each branch, to encode the different lottery probabilities.
(TIF)

**S1 Appendix. Analytical derivation of absorption probabilities.**
(DOCX)

## Author Contributions

**Conceptualization:** Giuseppe M. Ferro, Didier Sornette.

**Formal analysis:** Giuseppe M. Ferro, Didier Sornette.

**Investigation:** Giuseppe M. Ferro.

**Methodology:** Giuseppe M. Ferro.

**Project administration:** Didier Sornette.

**Resources:** Didier Sornette.

**Supervision:** Didier Sornette.

**Validation:** Giuseppe M. Ferro.

**Writing – original draft:** Giuseppe M. Ferro.

**Writing – review & editing:** Didier Sornette.

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
