## [Decision Letter · Decision Letter 0]

16 Sep 2020

PONE-D-20-19208

Stochastic representation decision theory: How probabilities and values are entangled dual characteristics in cognitive processes

PLOS ONE

Dear Dr. Ferro,

Thank you for submitting your manuscript to PLOS ONE. After careful consideration, we feel that it has merit but does not fully meet PLOS ONE’s publication criteria as it currently stands. Therefore, we invite you to submit a revised version of the manuscript that addresses the points raised during the review process.

As you can see from the report, the reviewer sees merit in the paper but has some concerns.  Please specifically address those concerns in any revision you submit. 

We look forward to receiving your revised manuscript.

Kind regards,

Jason Anthony Aimone

Academic Editor

PLOS ONE

Journal Requirements:

Reviewers' comments:

Reviewer's Responses to Questions

**Comments to the Author**

1. Is the manuscript technically sound, and do the data support the conclusions?

Reviewer #1: Partly

2. Has the statistical analysis been performed appropriately and rigorously? 

Reviewer #1: N/A

3. Have the authors made all data underlying the findings in their manuscript fully available?

Reviewer #1: Yes

4. Is the manuscript presented in an intelligible fashion and written in standard English?

Reviewer #1: Yes

5. Review Comments to the Author

Reviewer #1: I've certainly not seen a model quite like this before. There are DDMs for more than two options, but I've not seen a model that gives an absorbing state for each outcome. And the non-linear interaction of probabilities and utilities is novel. The motivation for this particular non-linear interaction is a bit unclear, but the model could certainly be adjusted if desired.

The description of the time-constrained model is a bit terse. Should I interpret the probability of L1 being chosen as the overall probability that the particle is on one of L1's outcomes' paths when time expires (t=T)?

The function f(t), which determines which branch the Brownian particle begins, seems fairly critical, but I didn't understand its derivation.

Why is probability perception defined as the relative probability of that event being the absorbing state if the lottery is chosen? How could this be observed, when we only can observe which lottery is chosen and not which outcome was absorbing? Normally we think of eliciting beliefs via bets (as described by Savage); is there a similar way we could elicit beliefs under this model?

More generally, with enough data, can we separately identify the utility and belief functions? Can I take data, estimate these functions, and use the estimates to make predictions?

The inverse-S shape of pi(p) is interesting. Is there evidence that changing lottery payoffs changes estimated inflection points? But also, previous observations of an inverse-S curve were derived assuming a linear combination of probability weights and utility weights; if your model is instead the right model would that past data actually result in an inverse-S pattern in estimated probability weights? It's not clear that your model should replicate features of other models that are structurally quite different.

Similarly, why apply a CARA utility function? Within the context of your model, does that functional form have particular appeal or empirical validity? Since your model is not an expected utility model, should we really interpret concavity of U(o) as risk aversion? Perhaps there are other properties of your model that would more properly characterize risk aversion. I find the analysis of choice probabilities to be more insightful.

The analysis of "evident" dominance is a bit unsatisfactory; simply showing that the dominant lottery is chosen more frequently still leaves open the chance that the dominated option is chosen reasonably often. Perhaps some comparative statics would be useful here.

Equation 28 may have a typo: B instead of L2. Furthermore, the relationship in that equation doesn't necessarily imply that harder decisions take more time. It could be that as P(L1) and P(L2) approach 0.5, E[T|L1] and E[T|L2] both approach 0 or both approach infinity. Equation 28 only governs their ranking, not their magnitude.

I'm confused by the section on double-S probability weighting. The Kahneman and Tversky (K-T) editing phase and the Al-Nowaihi and Dhami (AN-D) model predict a inverse double-S curve, while your model predicts that the curve changes from inverse-S- to S-shaped as time pressure changes. But for no value of T do you observe an inverse double-S curve, and, as far as I know, the K-T and AN-D results were derived without varying time pressure.

6. PLOS authors have the option to publish the peer review history of their article (what does this mean?). If published, this will include your full peer review and any attached files.

Reviewer #1: No

---

## [Author Response · Author response to Decision Letter 0]

10 Oct 2020

Dear Editor,

We are grateful to the reviewer for the generous comments. We have edited the manuscript to address his/her concerns.

We believe that the manuscript is now suitable for publication in PLOS ONE.

Giuseppe M. Ferro (PhD candidate at ETH Zurich)

On behalf of all authors.

 The description of the time-constrained model is a bit terse. Should I interpret the probability of L1 being chosen as the overall probability that the particle is on one of L1's outcomes' paths when time expires (t=T)?

The time-constrained absorption probability is the probability to be absorbed by one of the outcomes of L1, given that the particle is absorbed somewhere before time T. We incorporate this comment in Section “Stage 2: Finite time SRDT”.

 The function f(t), which determines which branch the Brownian particle begins, seems fairly critical, but I didn't understand its derivation

Eq. (4) is just one possible way to look at the problem, i.e. solving a diffusion process on each branch independently and then matching the flux to ensure conservation of probability mass. However, we did not proceed this way: rather, we first solve the absorption problem in the case of only two branches (i.e. a one dimensional Brownian motion between two absorbing walls), and then show how it can be generalized to an arbitrary number of branches. Following this approach the function needs not to be identified. We added a comment after Eq. (4) to stress this aspect.

 Why is probability perception defined as the relative probability of that event being the absorbing state if the lottery is chosen? How could this be observed, when we only can observe which lottery is chosen and not which outcome was absorbing? Normally we think of eliciting beliefs via bets (as described by Savage); is there a similar way we could elicit beliefs under this model? More generally, with enough data, can we separately identify the utility and belief functions? Can I take data, estimate these functions, and use the estimates to make predictions?

As correctly pointed out, our definition of subjective probability is rather different from the existing literature. Mathematically, its definition is reasonable because it obeys the correct limit π(p)→p as D→∞. However, we would like to stress that our mathematical formulation of the subjective probability is fundamentally dissimilar from that in Expected utility theories, where the separation between preferences and beliefs allows one to separately identify the subjective probability. In our framework, the subjective probability is actually implied by the utility function, and therefore two “separate” functions cannot be really identified. Our definition of subjective probability is a way to extract how the choice probability depends on the outcome probabilities, and to get an intuition on why our model is able to explain the fourfold pattern of risk preferences. Concretely, in our model, one would just need to estimate the utility function together with the parameters D and T (through e.g. maximum likelihood estimation) and the corresponding “belief function” comes as result. Because of this, we do not face the joint hypothesis testing issue when trying to simultaneously estimate utility and subjective probability. We have added this comment at the end of subsection “Subjective Probability without time-constraints”.

 The inverse-S shape of pi(p) is interesting. Is there evidence that changing lottery payoffs changes estimated inflection points? But also, previous observations of an inverse-S curve were derived assuming a linear combination of probability weights and utility weights; if your model is instead the right model would that past data actually result in an inverse-S pattern in estimated probability weights? It's not clear that your model should replicate features of other models that are structurally quite different.

There is evidence that changing lottery payoffs changes inflection points: in [1], for each individual, the authors perform an elicitation of two weighting functions π_S^- (p) and π_L^- (p) for gambles involving small. On average, they find that small probabilities (≤0.33 for small losses and ≤0.5 for large losses on mean data) are overweighted (indicating pessimism). The usual inverse-S shape thus holds over both small and large losses, but the inflection point shifts to the right over large losses. We have added this piece of evidence in subsection “Subjective Probability without time-constraints”.

It is true that our model should not blindly replicate features of other models, like CPT. We have focused more on the analysis of the choice probabilities, adding a new Section “Fourfold pattern of risk preferences”, where we show how our theory correctly predicts the fourfold pattern of risk (FFP) preferences at the level of choice probabilities. As also explained in point 3, our definition of subjective probability is a way to extract the dependence of choice probability on outcome probabilities. The results on subjective probability and utility function are one way to interpret how these two entities interact in our model, despite their mathematical non-separability. 

In other words, if in CPT the FFP is explained through the combination of concave-convex value function and non-linear probability weighting, we show that also in our framework it is possible to identify these constructs and they provide similar predictions.

 Similarly, why apply a CARA utility function? Within the context of your model, does that functional form have particular appeal or empirical validity? Since your model is not an expected utility model, should we really interpret concavity of U(o) as risk aversion? Perhaps there are other properties of your model that would more properly characterize risk aversion. I find the analysis of choice probabilities to be more insightful.

We agree. Within our model, one cannot tell a priori that the CARA function has some empirical validity. However, also in our theory, greater concavity implies more risk-aversion, as we now show in Fig 3 and 4. We also stress in Section “Fourfold pattern of risk preferences” that the coefficient of absolute risk-aversion r has no “absolute” meaning in our theory, in the sense that the risk attitudes of the individual ultimately depend also on the other parameters, and . On this aspect, we now show in Fig 6 that is an important parameter characterizing risk-aversion, as also discussed in subsection “Preference reversal with time pressure”. 

More generally, we stick with the notion of (initially concave) utility for the following reason: utility is a well-defined concept in choice under certainty [2], where diminishing marginal utility indicates less and less increase in “happiness” as wealth increases. In Expected Utility, the concept of diminishing marginal utility and risk-aversion are fundamentally entangled: there cannot be one without the other. On the other hand, in generalized theories like Rank-Dependent Utility Theory, as shown in [3], it is possible for a decision-maker to be risk-seeking with a concave utility function, provided that the probability weighting is sufficiently “optimistic”. Therefore, the concept of diminishing marginal utility and risk-aversion are decoupled to same extent. Analogously, our model hypothesis is that the utility function, when in the context of choice under certainty, has some form (e.g. the CARA function used in the manuscript), expressing (or not) diminishing marginal utility. Then, as soon as there is some uncertainty, due to the interaction between probability and value, the utility becomes “distorted”, and assumes a form like the one in Eq. (25), allowing to exhibit risk-seeking behavior for losses. We have added this comment at the end of subsection “Probability-distorted effective utility”.

 The analysis of "evident" dominance is a bit unsatisfactory; simply showing that the dominant lottery is chosen more frequently still leaves open the chance that the dominated option is chosen reasonably often. Perhaps some comparative statics would be useful here.

In Fig 12, we now show the probability of choosing the dominant option in Eq. (31) as a function of the parameters and . For small values of both parameters, the choice becomes more deterministic. More generally, as discussed in previous points, we have expanded significantly our analysis on choice probabilities: in Section “Fourfold pattern of risk preferences”, we investigate how the different parameters characterize risk attitudes. 

 Equation 28 may have a typo: B instead of L2. Furthermore, the relationship in that equation doesn't necessarily imply that harder decisions take more time. It could be that as P(L1) and P(L2) approach 0.5, E[T|L1] and E[T|L2] both approach 0 or both approach infinity. Equation 28 only governs their ranking, not their magnitude.

True, we understand that Eq. (28) does not imply that harder choices take more time, so we removed it. The general observed inverse relationship between choice probability and response time is now shown in Figure 13 (data from [4]) and conceptually described by Eq. (29). 

 I'm confused by the section on double-S probability weighting. The Kahneman and Tversky (K-T) editing phase and the Al-Nowaihi and Dhami (AN-D) model predict a inverse double-S curve, while your model predicts that the curve changes from inverse-S- to S-shaped as time pressure changes. But for no value of T do you observe an inverse double-S curve, and, as far as I know, the K-T and AN-D results were derived without varying time pressure.

In Fig. 8, fixing , we show that as decreases (time pressure increases), the probability weighting function changes from inverse-S to S-shaped. However, during this transition, for some values of (in the example ), we do observe an inverse double-S curve. The time constraint in our model is not necessarily meant as an external time pressure, but it can also be conceived as an internal time pressure, because of energy constraints and efficiency-accuracy tradeoff. Therefore, at this stage, we are not claiming that an explicit time-pressure is needed to recover the K-T and AN-D results. We have added this comment at the end of subsection “Realistic Inverse double-S-shaped probability weighting function”.

References

[1] Etchart-Vincent, N. (2004). Is probability weighting sensitive to the magnitude of consequences? An experimental investigation on losses. Journal of Risk and Uncertainty, 28(3), 217-235.

[2]Arrow, K. J. (1984). Individual choice under certainty and uncertainty (Vol. 3). Harvard University Press.

[3] Chateauneuf, A., & Cohen, M. (1994). Risk seeking with diminishing marginal utility in a non-expected utility model. Journal of Risk and Uncertainty, 9(1), 77-91.

[4] Petrusic, W. M., & Jamieson, D. G. (1978). Relation between probability of preferential choice and time to choose changes with practice. Journal of Experimental Psychology: Human Perception and Performance, 4(3), 471.

---

## [Decision Letter · Decision Letter 1]

25 Nov 2020

Stochastic representation decision theory: How probabilities and values are entangled dual characteristics in cognitive processes

PONE-D-20-19208R1

Dear Dr. Ferro,

We’re pleased to inform you that based upon the expert reviewer's evaluation, your manuscript has been judged scientifically suitable for publication and will be formally accepted for publication once it meets all outstanding technical requirements.

Kind regards,

Jason Anthony Aimone

Academic Editor

PLOS ONE

Additional Editor Comments (optional):

Reviewers' comments:

Reviewer's Responses to Questions

**Comments to the Author**

1. If the authors have adequately addressed your comments raised in a previous round of review and you feel that this manuscript is now acceptable for publication, you may indicate that here to bypass the “Comments to the Author” section, enter your conflict of interest statement in the “Confidential to Editor” section, and submit your "Accept" recommendation.

Reviewer #1: All comments have been addressed

2. Is the manuscript technically sound, and do the data support the conclusions?

Reviewer #1: Yes

3. Has the statistical analysis been performed appropriately and rigorously? 

Reviewer #1: Yes

4. Have the authors made all data underlying the findings in their manuscript fully available?

Reviewer #1: Yes

5. Is the manuscript presented in an intelligible fashion and written in standard English?

Reviewer #1: Yes

6. Review Comments to the Author

Reviewer #1: (No Response)

7. PLOS authors have the option to publish the peer review history of their article (what does this mean?). If published, this will include your full peer review and any attached files.

Reviewer #1: No

---

## [Editor Report · Acceptance letter]

27 Nov 2020

PONE-D-20-19208R1 

Stochastic representation decision theory: How probabilities and values are entangled dual characteristics in cognitive processes 

Dear Dr. Ferro:

I'm pleased to inform you that your manuscript has been deemed suitable for publication in PLOS ONE. Congratulations! Your manuscript is now with our production department. 

Kind regards, 

on behalf of

Dr. Jason Anthony Aimone 

Academic Editor

PLOS ONE